# Effects of Straw Mulching and Reduced Tillage on Crop Production and Environment: A Review

Changliang Du [1,2], Lingling Li [1,2,*] and Zechariah Effah [1,2]

1    College of Agronomy, Gansu Agricultural University, Lanzhou 730070, China
2    State Key Laboratory of Aridland Crop Science, Gansu Agricultural University, Lanzhou 730070, China
*    Correspondence: lill@gsau.edu.cn; Tel.: +86-0931-7631145

**Abstract:** Taking sustainable agriculture measures is critical to effectively cope with the effect of the increasing population on water shortage. Straw mulching and reduced tillage are the most successful measures adopted in arid and semi-arid regions which affect crop production by changing the crop environment. This review focuses on the effects of tillage and mulching on the soil environment, including soil organic matter, soil moisture, soil temperature, soil microorganisms, soil enzyme activity, soil fertility, soil carbon emissions, pests, weeds, and soil erosion. In addition, water use efficiency and crop production are discussed under different tillage measures. Straw mulching can increase soil organic matter content, adjust soil moisture, and prevent water loss and drought; however, it can also lead to an increase in pests and diseases, and change the structure of the soil microbial community. Straw mulching can significantly enhance WUE (water use effectively) and yield. Reducing tillage maintains soil integrity, which is conducive to soil and water conservation, but could negatively impact crop yield and WUE. Precise field management measures, taken according to crop varieties and local conditions, not only ensure the high yield of crops but also protect the environment.

**Keywords:** straw mulching; reduced tillage; crop yield; water use efficiency; environment





## 1. Introduction

The growing population has increased the competition for land, water, and other resources, whilst also raising the demand for food [1,2]. Arid and semi-arid regions account for about half of the world's total land area [3] and could play a vital role in solving the world's food security problems [4]. Increasing the yield per unit of grain is an effective measure to ensure food security. As agricultural production from drylands is limited due to the scarcity of rainfall, therefore, it is essential to increase crop productivity and yields by optimizing agricultural field management practices [5].

Adopting appropriate agricultural measures not only improves water use efficiency by minimizing the non-productive water consumption of farmland water but also increases productivity, reduces soil erosion, and improves the ecological environment [6]. Meanwhile, it is also necessary to eliminate the enormous pressure that agriculture occupies environmental resources. It is very important to choose field management measures that not only ensure the stability and high yield of agricultural production but also contribute to the sustainable development of the ecological environment. Therefore, it is vital to discuss the effects of field management measures on the crop yield and soil environment associated with crop plants.

Among different agricultural measures, mulching, tillage, and their combination are considered as most sustainable agricultural practices in arid and semi-arid regions as they retain water in the upper soil layers reducing the need for irrigation [7]. Compared with traditional soil tillage measures, the implementation of mulching and reduced tillage techniques can significantly reduce soil surface water evaporation [8], surface runoff [9,10], and

soil erosion [11], and increase soil water storage. To an extent, it increases crop yield and water use efficiency and guarantees the sustainable development of agriculture [12,13]. Straw mulching has been widely used for the cultivation of maize (*Zea mays* L.) [14], spring wheat (*Triticum aestivum* L.) [15], rice (*Oryza sativa* L.) [16], potatoes (*Solanum, tuberosum*, L)[17], vegetables [18] and fruit trees [19]. This farming method not only guarantees the production of crops [20] but is also extremely friendly to the environment [21].

Straw mulching has many advantages: firstly, it reduces the loss of soil moisture and can be an effective means for improving plant available water [22], which in turn improves WUE and crop production [8,23]. Secondly, it balances the soil temperature [24] and reduces the maximum soil temperature but increases the minimum soil temperature [25]. Thirdly, it affects the soil microbial biomass [26]. Studies have shown that mulching significantly increases microbial biomass [27], but some contrasting studies have shown that conventional tillage can increase soil microbial biomass [28]. Fourth, it can affect the balance of soil organic carbon. A positive correlation has been found between organic residues in soil and soil organic carbon content [29,30]. Fifth, it can promote soil enzyme activity. The enzyme activity and metabolic index of straw mulching treatment were higher than those of conventional tillage [31]. Furthermore, straw mulching can suppress and reduce weed [32], as well as decrease runoff volumes [10].

Long-term tillage can affect soil function and thus crop yield [33]. Therefore, no-tillage or reduced tillage has been adopted in many areas of the world as field management to improve grain yield [33,34]. These measures play an important role in field management to improve grain yield. First, it can improve the level of soil organic matter which might offset the negative impact of residue removal [35,36]. Second, no-tillage or reduced tillage reduces the costs of production, increases soil compaction and bearing capacity, and can also alleviate the damage of wheel pressure [37] and increase area performance [38]. Third, no-tillage or reduce tillage can improve moisture infiltration [39] and enhance water use efficiency [40].

The tillage practices and mulching affect crop growth in slow processes by affecting the soil environment [41]. Thus, to ensure ecological stability and high yield, it is vital to review the effects of these measures on the crop soil environment. However, under field conditions, multiple tillage methods are often adopted. So it is difficult to understand the effects of a single tillage measure. In this review, we will discuss the effects of mulch and no-tillage or reduced tillage on the crop yield and environment.

## 2. The Effect of Straw Mulching and Reduced Tillage on the Soil Environment

### 2.1. Soil Organic Matter

Soil organic matter (SOM) plays a vital role in crop production by providing the most basic energy substrate and restoring degraded soils [42]. The decomposition of organic mulching by soil microorganisms produces high content of SOM, increases soil biodiversity, and maintains good ecosystem functions [43]. Proper mulching and irrigation have successfully increased the crop production of some farmlands with initial low soil fertility [44]. A long-term zero tillage with residues incorporation for 11 years showed a significant increase in the SOM and improved aggregate size distribution and stability [45]. Similar studies showed that the SOM increased significantly after straw mulching [46] and alfalfa and bark mulching [47].

Most studies have shown that no-tillage can increase the amount of SOM in the 0–100 cm topsoil [48,49] by reducing soil disturbance, mineralization, and decomposition of organic matter [50]. Baan et al. [51] reported that compared with long-term continuous tillage, a short cycle of single tillage had little effect on soil physicochemical properties. In addition, no or reduced tillage was effective in increasing free unprotected organic matter [52]. Wang et al. (2008) showed a 22% increase in the SOM content in the 0–100 cm soil layer after the long-term no-tillage compared with traditional tillage in the semi-arid Loess Plateau [53]. However, the effect of no-tillage on the SOM content is not well observed in arid and semi-arid agricultural areas showing some contradictory effects because of

the different soil environments and climatic conditions. In general, straw mulching with reduced or no-tillage has significant effects on improving soil organic matter.

### 2.2. Soil Moisture

Soil moisture directly affects the growth of plants, and reasonable soil moisture is the most important factor in ensuring high quality and high yield of crops. In arid and semi-arid areas, conserving soil moisture and increasing WUE are key to crop production [54]. Straw mulching reduces the evaporation of the soil moisture and water is available to plant roots in the upper soil layer which reduces evapotranspiration loss of underground water by capillary fringe [55]. In arid and semi-arid areas, straw mulching reduces evaporation during years with little rain, increases the movement of soil moisture [56], and improves the soil moisture at the depth of 0–40 cm compared with traditional tillage [57,58]. Similarly, in the Mediterranean environment, straw mulching significantly improved the soil moisture at the depth of 5–15 cm [59]. Several field studies had reported a positive impact of the combined application of no-tillage and reduced tillage on soil moisture which ensures the absorption and utilization of moisture by plants [60]. Furthermore, the straw left on the soil surface can minimize the negative impact of no-tillage on water infiltration [61].

### 2.3. Soil Temperature

Soil temperature generally refers to the temperature of the soil in the root growth layer and is related to the root growth. Soil temperature is generally affected by weather, soil moisture, topography, and farming conditions. The easiest way to control soil temperature is by changing field management, including mulching and tillage measures. Bare soils generally have higher temperature changes than covered soils due to albedo effects [62]. Covering soil reduces moisture exchange between soil and air, thereby reducing heat flow and exchange between the soil and air [62]. It is generally believed that farmland mulch works as a partial heat barrier that can prevent ultraviolet rays and reduce heat loss.

Most studies have concluded that straw mulching can reduce the surface soil temperature of spring wheat, corn, rice, and other root crops [55]. Straw mulching can decrease the upper limit of soil temperature and lower the maximum soil temperature, straw mulching may balance the soil's temperature and make it more favorable for plants to grow and thrive [62]. A study showed that crop residue covering significantly reduces the early plant growth temperature of 2–7 °C [63]. Although the no-tillage straw mulch reduces the surface temperature of the soil, it also increases the soil moisture and water use efficiency at the same time, thereby increasing yield [64]. Another study on rice showed that straw mulching significantly reduced the surface temperature when the temperature increased [56]. Similarly, with the use of wheat straw mulching to cover the plots for planting corn, the maximum soil temperature during the corn growing period lowered when the ambient temperature was high, and the soil temperature increased when the ambient temperature was low and promoted corn production [65]. The temperature of the soil surface improved with the thickness of the straw mulch layer. However, the temperature change mainly appears at 0–40 cm depth, and no significant effect was observed in deeper soil layers [66].

The ground temperature is affected by many factors such as climate, region, vegetation type, cover type, and cover time. It can be concluded that straw mulching and no-tillage practices can affect crop production by affecting soil temperature. It can be seen that the systematic study of soil temperature and its influencing factors have an important impact on the in-depth study of plant growth and development.

### 2.4. Soil Microorganisms

Soil microorganisms play a remarkable role in sustaining soil structure [67,68], humus formation [69], promote the circulation and flow of material energy in the soil [70], and simultaneously affect plant growth [71].

Straw mulching and less or no tillage could improve the soil microorganisms to varying degrees [70]. In a field study in India, Subrahmaniyan et al. [72] found that the numbers of soil bacteria, fungi, and actinomycetes increased by 2%, 12%, and 12%, respectively, under mulch conditions compared to no mulch conditions. Some studies have proven that the straw mulching had significantly increased the soil microbial biomass by 42% [73] and showed a significant increase in microbial biomass in the 1–9 cm soil layer [74]. However, some studies have proven that mulching has no significant effect on soil microorganisms [75]. Wang et al. have also proven that soil with no mulching has more microbial biomass [76].

Less or no tillage could define the microbial biomass indirectly by affecting soil moisture and temperature [77]. In the no-till system, the activity and quantity of soil microorganisms were significantly higher than those of the tillage soil, and the soil quality parameters of the topsoil microorganisms and the tillage frequency were negatively correlated [78].

Long-term less or no-till combined with straw mulching is more conducive to increasing soil microbial biomass [79]. Reducing soil tillage combined with straw mulching can increase the total amount of bacteria in the topsoil and increase the diversity of soil microorganisms [80]. Studies have shown a more beneficial rhizosphere microbial community under no-tillage which in turn increases crop growth and yield [81]. Some researchers reported no significant change in microbial biomass under no-tillage soil with other treatments [82]. However, the soil microbial biomass and activity is a complex topic that could be affected by the mulch composition and tillage depth determination [83].

### 2.5. Soil Enzymes

The transformation of soil nutrients and their absorption and utilization by plants is affected by soil enzymes [84]. Soil enzymes are important indicators in measuring soil nutrient status and nutrient cycling [85]. Straw mulching has been shown to improve soil enzymatic activity [86]. One study showed that the soil enzyme activities were significantly improved after 4 years of wheat straw mulching, compared with other treatments [86]. Meanwhile, a study showed that the soil enzyme activity was significantly correlated with the degree of compaction after straw mulching [87]. Straw mulching had shown a significant increase in the activities of protease, urease, sucrase, and alkaline phosphatase during rice and wheat cultivation [88] and white clover and yellow peas cultivation [89]. Crop residue mulch could improve Arylamidase activity compared to bare plots [90]. However, no significant difference was found in the activities of acid phosphatase, protease, and aryl sulfatase after red clover mulching [91].

Soil enzyme activities were also found closely related to the tillage system. This might be because tillage practices can affect the stratified distribution of soil organic matter, thereby affecting the growth of soil microorganisms, which in turn affects the activity of soil enzymes [92] and soil nutrient content [93]. The enzyme activity of the subsoiling mulching treatment was higher than the no-till mulching treatment [94]. Some researchers have also shown that the different physical and chemical properties of soil from different places could affect the distribution and activity of soil enzymes [95].

### 2.6. Soil Fertility

Straw mulching and tillage either indirectly or directly improve soil fertility. Studies have shown that straw mulching not only increases soil temperature and moisture but also improves soil microbial biomass and nutrient cycling, thereby increasing the content of rhizosphere mineral nitrogen [96]. Straw mulching is itself a source of organic C and SOM which increase the availability of soil nutrients to plants and microbes [97]. Furthermore, it increased the soil's inorganic nitrogen and microbial carbon and nitrogen content [98] and improved soil physicochemical properties [99]. In shaded coffee agroecosystems, single mulching significantly increased C and N in the soil depth of 0–20 cm compared with no mulching [43]. The study also found that mulching significantly increased soil

exchangeable potassium and phosphorus compared to traditional rice-wheat cropping systems [100]. However, some researchers have reported no clear relationship between soil fertility and straw mulching [101]. No studies have reported the direct positive impact of land tillage practices on soil fertility; however, the combination of straw mulching and tillage practices significantly affects soil fertility.

*2.7. Soil Emissions*

In recent years, the interest of researchers has rapidly increased in evaluating the effects of different agroecosystems on $N_2O$ emissions. By comparing the mulching with the non-mulched treatment, it was found that the mulching could effectively reduce the $N_2O$ emission [102–104]. Similar studies also showed that the average $N_2O$ emissions from the uncovered treatments were higher than those of the covered treatments in the vicinity of the North Pacific [105]. However, some studies have also shown that surface mulching significantly enhances [106] or has no effect [107] on $N_2O$ emissions.

Compared with the tillage measures, no-tillage can reduce the concentration of soil electron receptors in paddy fields and inhibit $N_2O$ and $CH_4$ emissions [108]. No-tillage can usually increase soil bulk density and reduce soil porosity, resulting in lower emissions [109], but the amount of emission reduction varies due to differences in soil properties [87]. Low soil soluble organic and carbon content [110] and low soil temperature [111] under no-till may also be responsible for reducing emissions in the field. Van Kessel et al. [112] have shown that long-term (more than 10 years) conservation tillage can reduce $N_2O$ emissions in arid environments because of the better soil structure. However, other studies have also shown that no-till and conventional farming have no significant effects on soil emissions [113]. Some researchers believe that the anaerobic environment of no-till soil can promote denitrification, resulting in higher N leaching [114,115].

*2.8. Insect Pests, Weeds, and Soil Erosion*

Straw mulching has also been shown to play an important role in the distribution of pests in cultivation. For example, straw mulching increases insect abundance [68,116], meanwhile, it could decrease the insect damage in buckwheat and cabbage [117]. In shaded cannabis, sorghum, and sudangrass non-covered fields, the numbers of *Formicidae*, *Orthoptera,* and *Phyllanthus* were significantly higher than in covered fields [118]. It has been proven that straw mulching significantly changes the disease and pest spectrum [118,119]. Some studies believe that reducing tillage increases the bulk density of the topsoil and reduces the porosity of the soil, resulting in excessive soil nutrient enrichment on the surface aggravating pests and insect pests [120]. Studies have shown that straw mulching can reduce the population of *Myzus persicae* (*Aphididae*) on kale (*Brassicaceae*) plants [118]. Generally, straw mulching controls weed growth by limiting resources [121], especially straw mulching can significantly inhibit the growth of weeds [122]. The weed biomass was significantly lower in covered plots compared to bare plots [123], which was significantly reduced by straw mulch [124]. However, some studies have shown that straw mulching has no significant effect on aboveground biomass and weed numbers [125].

Mulching effectively controls soil erosion, especially in rain-fed areas [126]. Mulching techniques can increase soil moisture to stabilize topsoil and reduce wind erosion during periods of high evaporative capacity [127]. Mulching could reduce soil erosion by increasing infiltration [128], reducing the impact of raindrops [129], reducing runoff velocity [9], improving soil structure and porosity [130], and improving topsoil biodiversity [131]. No-tillage or reduced tillage measures can reduce soil erosion and nutrient loss [132], thereby preventing soil erosion [133]. Above all, the effect of straw mulching and tillage management on the soil environment is shown in Figure 1.

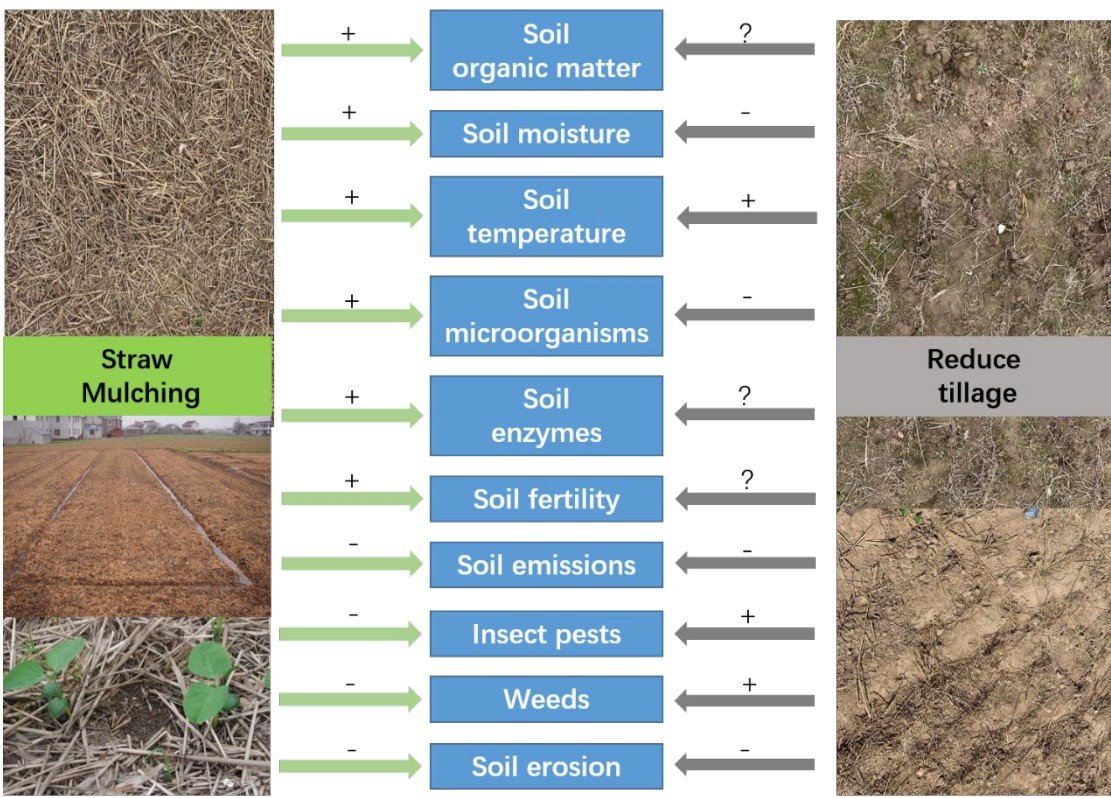

**Figure 1.** Effects of straw mulching and reduced tillage on the soil environment. The + sign showed a positive effect, the − sign showed the negative effect, and ? indicate the undefined effect.

## 3. Effects of Straw Mulching and Reduced Tillage on Crop Growth, Grain Yield, and WUE

### 3.1. Water Use Efficiency

Water use efficiency (WUE) usually refers to the ratio of grain yield or crop biomass to water consumption, which can reflect the growth and production of crops per unit of water consumed [134]. Straw mulching increases crop yield and water use efficiency. According to some articles, compared with conventional tillage, straw mulching can improve water use efficiency by 9–60% [135]. This may be because straw mulching can effectively inhibit the evaporation of soil water and increase the water available for the transpiration of crops [136]. Mulching increases the collection of natural water, which was transported through capillaries to low-lying areas for better uptake by crops [137–139]. Wang et al. [140] collected 1406 WUE values of straw mulching and no mulching field treatments and found that the water use efficiency of crops under mulching treatment was higher than that of no mulching treatment, among which wheat was the most prominent (Table 1).

**Table 1.** Effects of straw mulching and no-mulching with nitrogen application rate on water use efficiency. Adapted with permission from Ref. [140]. 2019, Wang, X.

|  | Mulching | | No-Mulching | |
|---|---|---|---|---|
| Rice | y = 0.035x + 5.0969 | $R^2$ = 0.4863 | y = 0.0318x + 4.924 | $R^2$ = 0.2788 |
| Maize | y = 0.0489x + 8.2747 | $R^2$ = 0.4304 | y = 0.0557x + 5.0969 | $R^2$ = 7.0112 |
| Wheat | y = 0.0635x + 2.8018 | $R^2$ = 0.6877 | y = 0.0352x + 6.2532 | $R^2$ = 0.4121 |

Studies have shown that no-tillage measures can effectively improve the water use efficiency of crops, especially under drought conditions [141]. Multiple studies in the Mediterranean region have concluded that no-till and low tillage not only increase soil moisture but also improve water use efficiency [142]. Other studies have also proved that

less tillage preserves more water in the topsoil layer compared to traditional farming [143]. However, contradictory studies reported that crop growth is more efficient in dry farming and deep tillage than in no-tillage implemented for several years [144]. In addition, some other studies show that the conservation tillage methods, such as no-tillage, do not affect soil water retention, WUE, or crop yield compared to traditional farming [145].

### 3.2. Grain Yield

Mulching can effectively change the growing environment of crops, by capturing and utilizing rainfall, which is effective in reducing the risk of crop failure in the field especially in arid or semi-arid regions [29,103]. For example, mulching can effectively increase soil organic matter content, reduce soil moisture evaporation, and improve soil temperature. Grain yield was higher under mulch than on bare land due to the significant effect of mulch in retaining soil moisture and increasing nutrient availability and transfer [146]. Compared with conventional tillage, the application of wheat straw mulch can improve maize yield and economic benefits [104]. Wang et al. [140], reported the effect of mulching on the yield data of 1516 samples of corn, wheat, and rice and showed that the average grain yield of the mulched treatment was higher than that of the no-mulched treatment to varying degrees (Table 2).

**Table 2.** Effects of straw mulching and no-mulching with nitrogen application rate on yield. Adapted with permission from Ref. [140]. 2019, Wang, X.

|  | Mulching | | No-Mulching | |
|---|---|---|---|---|
| Rice | y = 14.598x + 5437.9 | $R^2$ = 0.4729 | y = 13.585x + 4964.8 | $R^2$ = 0.4152 |
| Maize | y = 19.313x + 4481.6 | $R^2$ = 0.5481 | y = 18.736x + 3221.9 | $R^2$ = 0.5019 |
| Wheat | y = 15.334x + 2198.3 | $R^2$ = 0.4278 | y = 11.34x + 2514.2 | $R^2$ = 0.3435 |

However, different crops have different requirements for the soil temperature at different growth stages. Studies by some researchers have shown that the increase in soil temperature in the early stage of the crop caused by mulching promotes growth and development [147]. However, it has been reported that an increase in soil temperature in the later stages of growth can accelerate crop senescence and reduce crop dry matter accumulation, thereby reducing crop yield [148]. Some studies also showed that mulching and reduce tillage reduce onion yield and quality [149]. Similarly, in the arid and semi-arid regions of the Loess Plateau in Northwest China, the application of straw mulch is limited because straw mulching reduces soil surface temperature, resulting in decreased grain yields [140]. In low-lying and poorly drained fields, straw mulching can prevent rainfall from being drained in time, causing waterlogging and reducing yields [20].

Arvidsson et al. [150] found that no-till treatment of winter wheat yields was on average lower by 9.5% than that of tillage, probably because long-term no-till increases soil bulk density, hindering moisture connections and root growth between soil layers at different depths [151]. Studies have found that under drought conditions, no-tillage is conducive to the development of wheat [152]. Other studies have also shown that no-tillage can effectively improve the soil environment, resulting in improved yields [34,36]. No mulching and fewer tillage practices can also increase WUE and crop yields [153]. Ashworth et al. [154] studied the effects of no-till and conventional tillage on wheat yields, and their study showed that under no-tillage, higher yields were achieved compared to conventional tillage [50]. Long-term studies have shown that the continuous use of no-till varies greatly over different years [155]. Neugschwandtner et al. [156] reported that no-tillage corn yields were lower than conventionally cultivated maize in the first three years investigated; however, after three years the yields under no-tillage exceeded those of conventional ploughing. The increase in yield was significantly higher in the dry years than in the wet years [157]. After implementing no-tillage, positive soil promoting factors are expected to increase over time [158]. Analysis by Van Ittersum et al. [159] showed that

the initial application of no-tillage may have negative results at the start but long-term no-tillage has potential benefits such as reduced fertilizer demand and stable crop yields. The high yield of crops depends on high-quality soil biological and physicochemical properties. Especially in arid and semi-arid regions, straw mulching and no-tillage can ensure a good soil environment for crop growth, thus ensuring the sustainability of crop production.

## 4. Suggestions for Future Research

It is undeniable fact that less tillage, no-tillage, and straw mulching play an important role in agricultural production, especially in arid and semi-arid regions. These mainly include increasing soil organic matter, improving soil water temperature, reducing surface water evaporation, changing soil microbial structure, promoting soil enzyme activity, and inhibiting weed growth. In addition, crop straw mulching increases the collection and utilization of water, which is conducive to the prevention and control of soil erosion. At this stage, the problem of global warming is becoming more and more serious, resulting in climate change, which directly or indirectly affects agricultural production. The goal of agricultural success is sustainable development, protecting the environment as much as possible while maintaining agricultural field productivity, especially rain-fed agriculture in arid and semi-arid regions. In this direction, there will be a large number of topics to be researched in the future, but the main tasks we are currently facing are as follows.

### 4.1. Select a Combination of Agricultural Practices

In many parts of the world, especially in rain-fed farming regions, smallholder farmers make up the majority of the population, and it is a realistic requirement for them to develop Low-cost input, easy-to-operate, high-efficiency tillage measures. These include tillage, mulching, and fertilization. The combined application of various planting measures can ensure agricultural production and at the same time take good care of the environment; however, the optimal planting measures which are suitable for different regions in the world, including the optimal amount of fertilization, mulching materials, and tillage depth need further research.

### 4.2. Explore the Role of Trace Elements

Trace elements play a key role in maintaining crop growth. Some researchers believe that the trace elements enriched in crops are closely related to the planting environment. However, the specific mechanism of trace elements in plants and mulching or tillage is still unclear. Therefore, studying the relationship between plant trace elements and the environment is of great significance for us to better ensure food safety and environmental health.

### 4.3. Application of Plant Physiology in Agriculture

Various hormones and metabolites in plants play a vital role in plant growth. In crop plants, for example, when drought occurs, the roots send hormonal signals to the leaves, causing the stomata to close. This root-induced signal often helps reduce water loss, allowing photosynthesis to continue, thereby increasing water use efficiency. However, at this stage, the research in the direction of plant physiology is mainly carried out in the laboratory, which is separated from the actual agricultural production in the field. In field agriculture, the mechanism of plant physiology and biochemistry needs to be further studied, which is of great significance to study the crop growth.

### 4.4. Risk of Soil Pollution

No-tillage can cause soil compaction and insufficient soil fertility. At the same time, inappropriate straw mulching may increase the risk of pathogenic microorganisms and pest. Therefore, the maximum coverage area under straw mulching should be carefully investigated in combination with materials such as bio-based polymers and biodegradable polymers. Minimizing the use of pesticides minimizes their potential impact on the environment. Therefore, future research can access their impact on the environment.

## 5. Conclusions

Two common tillage measures i.e., straw mulching and tillage reduction in arid and semi-arid regions have an impact on the environment, including soil temperature and humidity, microorganisms, enzyme activities, soil erosion, and greenhouse gas emissions, as well as crop growth, production and water use efficiency. Straw mulching creates a physical barrier between the surface and the atmosphere and can significantly improve soil moisture evaporation and soil erosion, regulate soil temperature, and promote plant growth. Tillage measures have a certain impact on soil structure, and different tillage measures in different regions can promote agricultural production. In addition, straw mulching and reduced tillage directly affect the microenvironment of soil microorganisms, changing the sustainability of the environment. The rational utilization of tillage practices is an important challenge for crop production and requires more detailed studies to properly utilize resources under different environments and soil conditions.

**Author Contributions:** Conceptualization, L.L.; software, validation, L.L., C.D. and Z.E.; investigation, L.L.; resources, L.L.; data curation, C.D.; writing—original draft preparation, C.D.; writing—review and editing, L.L.; supervision, L.L. All authors have read and agreed to the published version of the manuscript.

**Funding:** This research was funded by the National Natural Science Foundation of China (31761143004); Education Science and Technology Innovation Project of Gansu Province (GSSYLXM-02).

**Institutional Review Board Statement:** All patients involved in this study gave their informed consent. Institutional review board approval of our hospital was obtained for this study.

**Informed Consent Statement:** Not applicable.

**Data Availability Statement:** Not applicable.

**Conflicts of Interest:** The authors have no competing interests that might be perceived to influence the results and discussion reported in this paper.

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
