# Peer review of "Effects of Straw Mulching and Reduced Tillage on Crop Production and Environment: A Review"

_water, doi:10.3390/w14162471_

Round 1

Reviewer 1 Report

1. Content of Introduction is short. Research background is missing.
2. What is the new contribution in your work
3. State research gap & explain reason for doing this research work
4. What is the ultimate goal of your work
5. Give citation wherever required
6. What is the limitations of your study?
7. Describe the innovation and goals of this research to differentiate your work from others.
8. Give necessary citation wherever required
9. Have author have used all the references in their manuscript ?
10. Conclusion need to be modify as per the work carried out
11. Abstract is vague
12. How the cropping pattern is decided??

Major revision

Author Response

We thank the reviewer for the valuable comments. Our response to the specific comments are given below.

Specific comments:

-1. Content of Introduction is short. Research background is missing.

Answer: We have clarified the research background of our study in the Introduction section in the revised manuscript and a lot of content has been added appropriately, as suggested.

-2. What is the new contribution in your work

Answer: In this article, we review the impact of two farming practices on the environment and crop production. On the one hand, the article summarizes the effects of the two farming measures on the environment including soil temperature and humidity, soil enzymes, soil microorganisms, soil and water conservation, greenhouse gas emissions, weeds and pests; on the other hand, the article summarizes the effects of the two farming measures on crops The effects of production and water use efficiency are discussed in depth. and propose future research directions. The details are shown in the conclusion section of the article.

-3. State research gap & explain reason for doing this research work

Answer: We have clarified the purpose and the research gap of our study in the abstract section in the revised manuscript, as suggested.

-4. What is the ultimate goal of your work

Answer: The first is to describe the impact of the two farming measures on crops and the environment, and to give the initial researchers in this direction the most preliminary understanding; the second is to provide new ideas for researchers in this direction to study farming measures.

-5. Give citation wherever required

Answer: We have cited many citation to support our claim in our review in the revised manuscript.

-6. What is the limitations of your study?

Answer: First, the author has read too few relevant documents, and the relevant content cannot be described comprehensively; second, the article only expounds two farming measures, straw mulching and tillage reduction, and is not comprehensive enough to describe more tillage measures in actual field production.

-7. Describe the innovation and goals of this research to differentiate your work from others.

Answer: Most of the previous researchers' research on farming methods only focused on one point of research and analysis, such as soil temperature or soil microorganisms. This study comprehensively considers the impact of two farming measures on crop growth and the environment, weighs crop growth and environmental sustainable development, and proposes possible future research directions. The innovations and goals of this work are described in the newly revised high school introduction section.

-8. Give necessary citation wherever required

Answer: We have cited many necessary citation to support our claim in our review in the revised manuscript.

-9. Have author have used all the references in their manuscript ?

Answer: In the latest revised manuscript, we confirm that all citations are used after verification.

-10. Conclusion need to be modify as per the work carried out

Answer: We have rewritten the conclusion section to capture the major findings of our study, implications of such findings and recommendations for future studies, as suggested.

-11. Abstract is vague

Answer: In the latest revised manuscript, we have rewritten the Abstract section.

-12. How the cropping pattern is decided?

Answer: The crop straw directly covering the field soil surface or the crop straw being rolled into the field soil is regarded as straw mulching; the soil depth of the field is regarded as the standard, and the shallower the layer, the less tillage.

Reviewer 2 Report

Effects of straw mulching and reduce tillage on crop production and environment ; A Review

Dear authors,

I have read this review paper with interest. Before, I further proceed this paper, I ran the similarity scores from well used Turnitin plagiarism software and found out that overall it has 22% which is slightly higher similarity score for a journal paper (which should be less than 20%). I hope, the authors can reduce that to 20%. I have attached the similarity report for your attention.

However, my concern to a particular paper is there.

Xiukang Wang, Junliang Fan, Yingying Xing, Guoce Xu, Haidong Wang, Jian Deng, Yanfeng Wang, Fucang Zhang, Peng Li, Zhanbin Li. "The Effects of Mulch and Nitrogen Fertilizer on the Soil Environment of Crop Plants" , Elsevier BV, 2018.

This particular paper was heavily used for your manuscript and the similarity scores are around 8% from this paper. Therefore, this is simply UNACCEPTABLE

The authors have to seriously think about this issue and rectify if the editor decides to move with your manuscript.

In addition, please see the following comments.

1. What is your intention to develop this review is not clearly showcased from abstract. What is the research gap?

2. Authors have a significant number of references in their introduction which is good. However, there is no clear connection from these papers to your review. You basically have references for every few sentences. Can you dig into these references and explain their work a bit at least. If not, it is like getting sentence by sentence from published papers.

3. Who owns the figure 1? I mean the photographs? If they are from authors, please state it.

4. Structuring for section 2 is good.

5. Table 1 and 2 are interesting. How do you get these linear variations? Can you justify the linearity? 

6. Table 2 is not in the correct way in the text.

7. Conclusions are very wage. Need to strengthen it. 

Author Response

We thank the reviewer for the valuable comments. Our response to the specific comments are given below.

Specific comments

-1.What is your intention to develop this review is not clearly showcased from abstract. What is the research gap?

Answer: We have clarified the purpose and the research gap of our study in the abstract section in the revised manuscript, as suggested.

-2.Authors have a significant number of references in their introduction which is good. However, there is no clear connection from these papers to your review. You basically have references for every few sentences. Can you dig into these references and explain their work a bit at least. If not, it is like getting sentence by sentence from published papers.

Answer: At the beginning of writing the paper, we made basic annotations on the collected literature. As a result, in the later writing process, some documents with little relevance were collected together. We have updated and revised parts of the literature in our newly submitted revised manuscript, explaining the work of the investigators in most of the cited references. Thanks again for your valuable advice.

-3.Who owns the figure 1? I mean the photographs? If they are from authors, please state it.

Answer: We have changed the image in the new revised manuscript and guarantee that all photos were taken by the author.

-4. Structuring for section 2 is good.

Answer: Thank you again for acknowledging the structure of the article.

-5.Table 1 and 2 are interesting. How do you get these linear variations? Can you justify the linearity? 

Answer: The data related to Tables 1 and 2 are all cited elsewhere, and we have added citations in the new revised manuscript.

-6. Table 2 is not in the correct way in the text.

Answer: We have reformatted the tables in the revised manuscript by removing dividing lines specially in Tables 2.

-7. Conclusions are very wage. Need to strengthen it. 

Answer: We have rewritten the conclusion section to capture the major findings of our study,implications of such findings and recommendations for future studies.

Round 2

Reviewer 1 Report

revision is appropriate

Author Response

We thank the reviewer for the valuable comments. We have made further changes to the law and vocabulary. Thank you very much for your valuable comments. The attachment is the latest revised paper, please refer to it.

Reviewer 2 Report

Authors revisions are acknowledged. However, I still have my concern on similarity scores.

I have attached the Turnitin report.

Author Response

We thank the reviewer for the valuable comments. We have revised the thesis based on your comments. We lowered the similarity score. Further revisions have been made to grammar and vocabulary. Thank you very much for your valuable comments. The attachment is the latest revised paper, please refer to it.
